# Efficient spin excitation via ultrafast damping-like torques in antiferromagnets

Christian Tzschaschel [1,3✉], Takuya Satoh [2] & Manfred Fiebig [1]

Damping effects form the core of many emerging concepts for high-speed spintronic applications. Important characteristics such as device switching times and magnetic domain-wall velocities depend critically on the damping rate. While the implications of spin damping for relaxation processes are intensively studied, damping effects during impulsive spin excitations are assumed to be negligible because of the shortness of the excitation process. Herein we show that, unlike in ferromagnets, ultrafast damping plays a crucial role in anti-ferromagnets because of their strongly elliptical spin precession. In time-resolved mea-surements, we find that ultrafast damping results in an immediate spin canting along the short precession axis. The interplay between antiferromagnetic exchange and magnetic anisotropy amplifies this canting by several orders of magnitude towards large-amplitude modulations of the antiferromagnetic order parameter. This leverage effect discloses a highly efficient route towards the ultrafast manipulation of magnetism in antiferromagnetic spintronics.

[1] Department of Materials, ETH Zurich, 8093 Zurich, Switzerland. [2] Department of Physics, Tokyo Institute of Technology, Tokyo 152-8551, Japan. [3] Present address: Department of Chemistry and Chemical Biology, Harvard University, Cambridge, MA 02138, USA. ✉email: ctzschaschel@fas.harvard.edu

Antiferromagnets (AFMs) are a promising material class for spintronic applications[1,2]. Their robustness against external magnetic fields, along with the possibility of picosecond antiferromagnetic switching, could help to substitute semiconductor-based information technologies with anti-ferromagnetic spintronic devices[3,4]. In fact, an electrically switched antiferromagnetic bit has been presented recently[5] and synthetic AFMs have substantially improved the speed and data density of racetrack memories[6]. Efficient manipulation of AFMs relies on spin currents acting as effective magnetic fields generating staggered (anti-) damping-like torques[7]. Such electrical control, however, has two disadvantages. On the one hand, the required current densities are in the range of ~$10^6 - 10^8$ A cm$^{-2}$, which involves a considerable amount of Joule heating. On the other hand, timescales accessible with electrical currents are typically too slow to explore the aforementioned picosecond dynamics. In contrast, optical laser pulses provide the time resolution required to track THz spin dynamics. The inverse Faraday effect (IFE), i.e. the optical generation of effective sub-picosecond magnetic field pulses with circularly polarised light, allows to impulsively create a spin canting in AFMs[8–15].

For ferromagnetic excitations, the magnetic anisotropy typically leads to a circular or only slightly elliptical spin precession[16]. In AFMs, however, the competition between magnetic anisotropy and the ≳100 times stronger exchange interaction results in strongly elliptical trajectories. An oscillating net magnetisation occurs along the minor axis of the ellipse, whereas the antiferromagnetic order parameter is modulated due to a collective in-phase spin canting along the major axis[15,17,18]. Therefore, the efficiency of impulsive spin excitations in antiferromagnets depends strongly on the direction of the initial spin canting. In particular, an impulsively generated net magnetisation in an AFM would be highly efficient in triggering large-amplitude changes of the antiferromagnetic order.

Antiferromagnetic spin dynamics are typically described by the Landau–Lifshitz–Gilbert equation[19]

$$\frac{d\mathbf{M}_i}{dt} = -\gamma\mu_0\mathbf{M}_i\times\mathbf{H}_i - \gamma\mu_0\frac{\alpha}{M_0}\mathbf{M}_i\times(\mathbf{M}_i\times\mathbf{H}_i), \qquad (1)$$

where $\mathbf{M}_i$ denotes the magnetisation of the $i$-th antiferromagnetic sublattice with magnitude $M_0$, $\mathbf{H}_i$ is the field acting on $\mathbf{M}_i$, and $\alpha$ is the Gilbert damping parameter. Further, $\gamma$ and $\mu_0$ are the gyromagnetic ratio and the permeability, respectively. The first term of Eq. (1) is commonly referred to as the field-like torque $\mathcal{T}_{FL}$, whereas the second term corresponds to the damping-like torque $\mathcal{T}_{DL}$[7]. For fully compensated AFMs excited by the IFE, $\mathcal{T}_{FL}$ modulates the antiferromagnetic order parameter, but does not induce a net magnetisation at $t = 0$ (see Supplementary Movie 1). Therefore, the impulsive generation of a net magnetisation by the effective magnetic field of the IFE can only occur via an ultrafast damping-like torque $\mathcal{T}_{DL}$ (see Supplementary Movie 2). So far, however, damping has been considered to describe only the spin-relaxation after the excitation[20]. The ultrafast damping-like torque of the IFE was neglected for the excitation process itself, based on the shortness of the interaction and the suspected smallness of the torque.

Here, we highlight the crucial role of the ultrafast damping-like torque during the excitation process for efficient impulsive spin excitations in AFMs. We detect such damping in the initial phase of a coherent spin precession of hexagonal YMnO$_3$ and HoMnO$_3$. We tune the spin damping strength via temperature variations and study the influence of magnetic order–order and order–disorder transitions on the damping. Our phenomenological theory considers both damping and magnetic anisotropy effects during the excitation, and thus develops and quantifies a consistent picture of impulsively induced coherent spin dynamics.

Strikingly, we find that the hitherto neglected ultrafast spin damping during the impulsive spin excitation can even provide the dominant spin excitation mechanism.

## Results

**Impulsive spin excitations in damped h-$R$MnO$_3$.** YMnO$_3$ orders antiferromagnetically at the Néel temperature $T_N$ ~ 73 K with a transition from $P6_3cm1'$ ($T > T_N$) to $P6_3'cm'$ ($T < T_N$) symmetry[21] (Fig. 1a). HoMnO$_3$ undergoes a $P6_3cm1' \rightarrow P6_3'c'm$ transition at $T_N$ ~ 75 K followed by a first-order spin-reorientation transition (SRT) to $P6_3'cm'$ at $T_{SR}$ ~ 35 K[22,23]. In a mean-field approach, the free energy density for both YMnO$_3$ and HoMnO$_3$ reads

$$\mathcal{F} = \lambda\sum_{\langle i,j\rangle}\mathbf{M}_i\cdot\mathbf{M}_j + D_z\sum_i M_{i,z}^2 + D_y\sum_i M_{i,y}^2 \\ + \Delta\sum_i M_{i,x}^2 M_{i,y}^2, \qquad (2)$$

where $\mathbf{M}_i$ denotes the magnetisation at site $i$ ($i = 1, 2, 3$). Furthermore $\lambda > 0$ and $D_z > 0$ parametrise the antiferromagnetic intra-plane exchange interaction and the easy-plane anisotropy, respectively. The inter-plane exchange is two orders of magnitude weaker than the intra-plane exchange; thus, we neglect it[24]. $D_y < 0$ stabilises the ground state as $P6_3'c'm$ ($\mathbf{M}_i$ then points along equivalent crystallographic $y$ axes), whereas $D_y > 0$ stabilises $P6_3'cm'$ (with $\mathbf{M}_i$ along equivalent crystallographic $x$ axes). $\Delta \geq 0$ is a weak fourth-order in-plane anisotropy, which becomes relevant at the SRT, where $D_y$ crosses zero. The three-sublattice antiferromagnet exhibits three optically excitable spin precessions called $X$, $Y$, and $Z$ mode[18,25], which relate to an oscillating net magnetisation along the $x$, $y$ or $z$ axes, respectively. We investigate the damping-like torque in relation to spin excitations via the IFE, which leaves us with the $Z$-mode

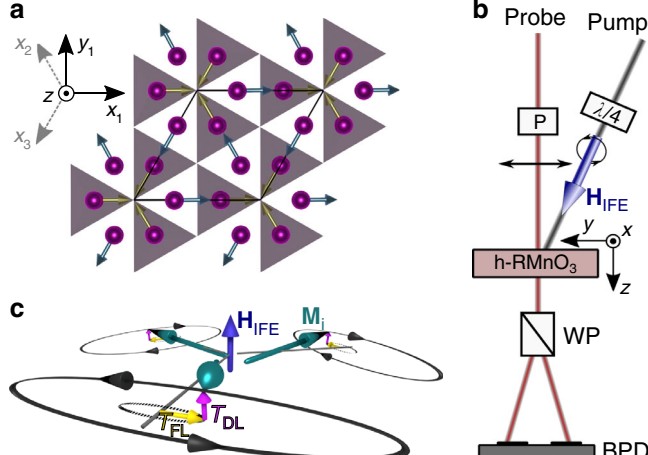

**Fig. 1 Experimental geometry. a** Projection of the YMnO$_3$ crystal structure onto the basal plane. Mn$^{3+}$ ions (violet) in grey and white areas are located in planes at $z = 0$ and $z = c/2$, respectively. The three sublattice magnetisations $\mathbf{M}_i$ point along equivalent $x$ axes $x_i$ as indicated in the coordinate system. Note that spins in HoMnO$_3$ point along equivalent $y$ axes for $T_{SR} < T < T_N$ (Supplementary Fig. 1). **b** Schematic of the setup. The pump and probe pulses are circularly and linearly polarised, respectively. P polariser, $\lambda/4$ quarter-wave plate, WP Wollaston prism, BPD balanced photodiode. **c** Visualisation of optical $Z$-mode excitation with field-like and damping-like torques $\mathcal{T}_{FL}$ and $\mathcal{T}_{DL}$ exerted by the effective field $\mathbf{H}_{IFE}$ of the IFE on the magnetisation $\mathbf{M}_i \parallel \hat{\mathbf{x}}_i$ ($\hat{\mathbf{x}}_i$ denotes the unit vector in the direction $x_i$). Black lines illustrate the ensuing strongly elliptical spin precession. Dashed ellipses show the expected trajectory without spin excitation via $\mathcal{T}_{DL}$. Precession amplitudes are significantly reduced.

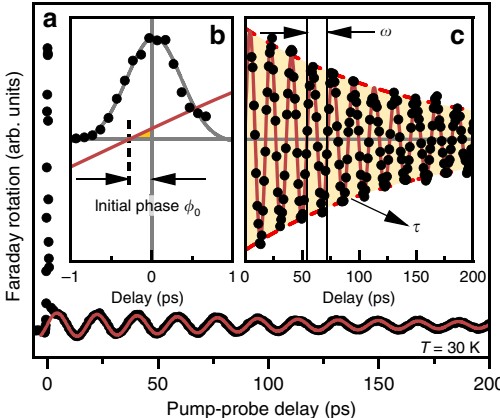

**Fig. 2 Z-mode precession in YMnO₃. a** Time-resolved Faraday rotation following the impulsive optical excitation at time $t = 0$. **b** Magnified view of the region around $t = 0$. **c** Offset-corrected oscillation (see Methods section). The red line is an exponentially damped sine-fit with the dashed line as its envelope. The extrapolation of the fit towards $t = 0$ is shown in **b** and reveals a finite deflection at $t = 0$. This initial deflection yields a temporal shift of the spin precession by ~280 fs, which amounts to an initial phase of 0.1 for a precession with a period of 18.2 ps.

excitation illustrated in Fig. 1c. Figure 2a shows an exemplary measurement of the time-resolved Faraday rotation in YMnO₃ after optical excitation with a circularly polarised pump pulse at 30 K. Two regimes can be distinguished: (i) a pronounced peak ~0 ps reflecting the direct interaction between pump and probe pulse and (ii) a damped sinusoidal modulation of the Faraday rotation that marks the ensuing spin precession and relaxation.

From a fit of the latter we extract the relaxation time $\tau$, frequency $\omega$, and initial phase $\phi_0$ of the precession (see Methods section). Figure 2b shows a magnified view of the region of the temporal overlap along with an extrapolation of the sinusoidal fit. Most strikingly, we find a finite magnetisation at 0 ps. This is unexpected because the conventional field-like torque of the IFE only perturbs the antiferromagnetic order, but does not induce a net magnetisation at the moment of the excitation[14,26–28]. We will show that the observed immediate onset of a finite net magnetisation is a consequence of spin damping during the impulsive spin excitation, which has been neglected so far.

Originally, YMnO₃ is in the magnetic ground state with $\mathbf{M}_i(t < 0) = M_0 \hat{\mathbf{x}}_i$, where $M_0$ is the sublattice saturation magnetisation. The optomagnetic field pulse of the IFE can be modelled as $\mathbf{H}_{\mathrm{IFE}}(t) = H\theta\delta(t)\hat{\mathbf{z}}$, where $H$ is the effective magnetic field strength, $\theta$ is the laser pulse duration, and $\delta(t)$ denotes the Dirac delta function[29]. Analytically integrating Eq. (1) from $t = -\infty$ to $+0$ with $\mathbf{H}_i = \mathbf{H}_{\mathrm{IFE}}$ yields:

$$\mathbf{M}_i(+0) = M_0 \begin{pmatrix} 1 \\ \gamma\mu_0 H\theta \\ \alpha\gamma\mu_0 H\theta \end{pmatrix}. \qquad (3)$$

The canting along the $z$ axis is typically significantly smaller than the $y$ canting because $\alpha \ll 1$. However, any spin canting along the magnetic hard axis ($z$) will be enhanced during the spin precession as a combined effect of antiferromagnetic exchange and magneto-crystalline anisotropy. This so-called exchange enhancement[1,14,17,29,30] is an effect that is intrinsic to AFMs, but is absent in ferromagnets. Thus, whereas damping effects during the spin excitation may be neglected for ferromagnets, this is not generally justified in AFMs. In fact, we will show that despite $\alpha \ll 1$, the damping-like contribution to the total spin excitation may even dominate over the field-like contribution.

The threefold rotational symmetry, which is preserved during the $Z$-mode precession, allows us to describe the antiferromagnetic dynamics by considering only one spin precessing in effective out-of-plane and in-plane anisotropies $\mathcal{J} = \frac{3}{2}\lambda + D_z + |D_y|$ and $\mathcal{D} = \Delta M_0^2 + |D_y|$, respectively (Supplementary Note 1). For the $z$ component of the oscillating net magnetisation $\mathbf{M}$, this yields a damped sinusoidal oscillation with

$$\omega = \omega_0 \sqrt{1 - A^2\alpha^2/4}, \qquad (4)$$

$$\tau = \frac{2}{A\alpha\omega_0}, \qquad (5)$$

$$\tan\phi_0 = A\alpha \frac{\sqrt{1 - A^2\alpha^2/4}}{1 - A^2\alpha^2/2}. \qquad (6)$$

Here, $\hbar\omega_0 = 2g\mu_B M_0\sqrt{\mathcal{JD}}$ is the precession frequency for the undamped case with the Landé factor $g = 2$ and the Bohr magneton $\mu_B$. $A = \sqrt{\mathcal{JD}^{-1}} \gg 1$ is the exchange enhancement factor[1,29]. Geometrically, $A$ corresponds to the aspect ratio of the elliptical spin precession[14,17]. Note that the initial phase $\phi_0$ depends on the damping during the impulsive spin excitation in direct relation to the initial phase in Fig. 2 and the ultrafast damping-like torque $\mathcal{T}_{\mathrm{DL}}$.

**Ultrafast spin damping in HoMnO₃.** We further investigate the role of the damping-like torque by performing temperature-dependent measurements of the optically induced spin dynamics. Temperature affects both the Gilbert damping parameter $\alpha$ (and therefore the magnitude of $\mathcal{T}_{\mathrm{DL}}$) and the magneto-crystalline anisotropy (and therefore the exchange enhancement $A$). In particular, we trace the $Z$ mode through the first-order SRT in HoMnO₃ and towards the second-order antiferromagnetic-paramagnetic phase transition in YMnO₃ (see Supplementary Fig. 2 for exemplary time-domain data). Conceptually, we thus investigate an order–order and order–disorder phase transition, thereby showcasing the general importance of spin damping during ultrafast antiferromagnetic spin excitations.

We first consider the case of HoMnO₃. Based on measurements as in Fig. 2, we extract $\omega_0$, $\tau$, and $\phi_0$ as depicted in Fig. 3a–c, respectively. The SRT causes a dip of $\omega_0$ at T~33 K. The fact that the frequency does not drop to zero at the SRT highlights the importance of the fourth-order anisotropy term in Eq. (2). All anisotropy parameters can be extracted from a fit of the temperature dependence of $\omega_0$ with a minimal model (see Methods section). (The deviations below 25 K are caused by the incipient ordering of the Ho(4b) moments[24,31,32], see Supplementary Fig. 3).

Moreover, using Eqs. (4) and (5), we determine the exchange-enhanced damping $A\alpha$ and, in combination with the extracted anisotropy constants, its constituents $A$ and $\alpha$ (Fig. 3d–f). We find a significant increase in $A\alpha$ between 30 K and 45 K. Together with Eq. (6) we can then calculate the temperature dependence of the initial phase and compare it with the direct measurement (Fig. 3c). The agreement around the SRT and towards higher temperatures is impressive and demonstrates the importance of the previously unrecognised damping-like torque $\mathcal{T}_{\mathrm{DL}}$. More strikingly, combining Eqs. (2) and (3) yields the energy density transferred into the magnetic lattice during the excitation as

$$\Delta\mathcal{F} = \mathcal{F}(+0) - \mathcal{F}(-\infty) \approx 3(A^2\alpha^2 + 1)\mathcal{D}\mathbf{M}_y^2, \qquad (7)$$

where the first term originates from the ultrafast damping-like torque along the $z$ axis, whereas the second term is related to

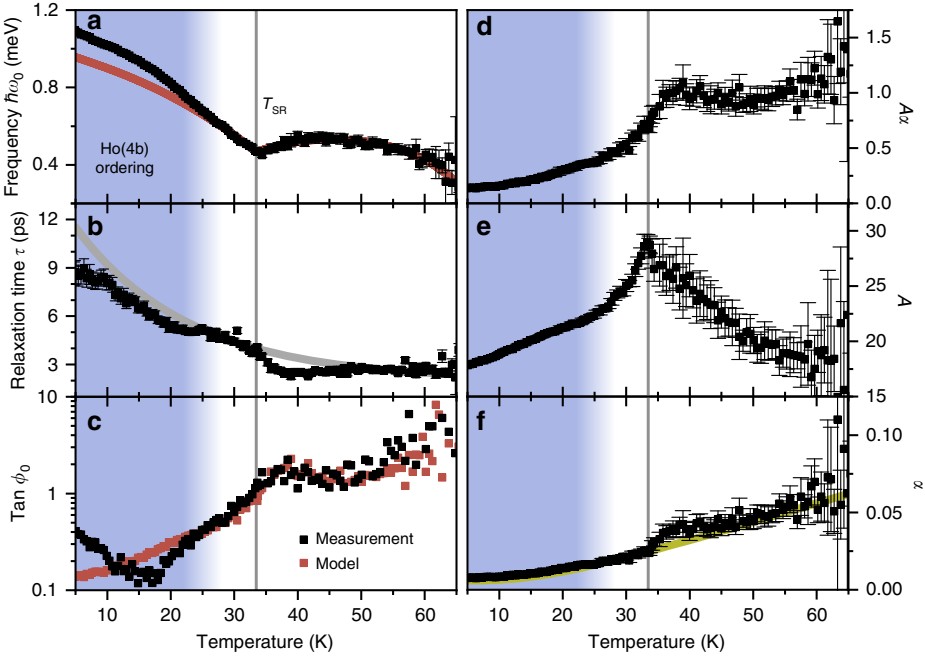

**Fig. 3 Temperature-dependent Z-mode characterisation in HoMnO₃.** **a** Frequency, **b** relaxation time, and **c** tangent of the initial phase of the optically induced spin precession as function of temperature. Red: model calculations (see Methods section). In the blue-shaded region ≲25 K, incipient ordering of the Ho(4b) moments leads to discrepancies between data and the model (see Supplementary Fig. 3). Grey line: exponential decay as guide to the eye. **d** Exchange-enhanced damping parameter $A\alpha$ extracted from Z-mode frequency and relaxation time (Eqs. (4) and (5)). **e** Temperature-dependent aspect ratio $A$ of the spin-precession ellipses. **f** Resulting Gilbert damping parameter $\alpha$. Yellow line: function proportional to Bose-Einstein distribution with quasiparticle energy of 4.3 meV.

the field-like in-plane torque of the IFE. Thus, as $A\alpha$ exceeds one, spin excitation via the damping-like torque of the IFE even becomes the dominant excitation mechanism. The enhancement of $A$ around the SRT is based on changes of the magneto-crystalline anisotropy, while the anomalous increase of $\alpha$ matches the observation of a temporary small-domain state in passing the SRT[23]. Such a small-domain state exhibits a high density of domain walls which will locally affect the spin precession frequency. Thus, we attribute the enhanced Gilbert damping to dephasing due to magnon scattering at the walls[33].

**Ultrafast spin damping in YMnO₃.** The general trend of $\alpha$ is captured well by an offset Bose-Einstein distribution for a quasiparticle with an energy of ~4.3 meV, suggesting that scattering with the crystal-field excitations of the Ho 4f electrons (≈3.5 meV[24]) contributes significantly to the spin damping. This is confirmed by measurements on YMnO₃ (Fig. 4), yielding relaxation times that are ~20 times longer than those for HoMnO₃, in line with the unoccupied 4f orbital. Accordingly, the extracted values for the temperature-dependent Gilbert damping parameter $\alpha$ in YMnO₃ (Fig. 5) are an order of magnitude lower than those in HoMnO₃. At low temperature, $\alpha$ can be approximated by a Bose-Einstein distribution with a quasiparticle energy of 2.6 meV, which corresponds to a hybrid spin-lattice mode in YMnO₃[34]. Assuming that the frequency of this mode is, just like the Z-mode frequency, proportional to the magnitude of the antiferromagnetic order parameter, we can consistently describe the temperature dependence up to the Néel temperature.

Note, however, that despite good qualitative agreement, Fig. 4c reveals a difference between the measured and modelled value of the initial phase $\phi_0$. This indicates secondary, non-damping-based

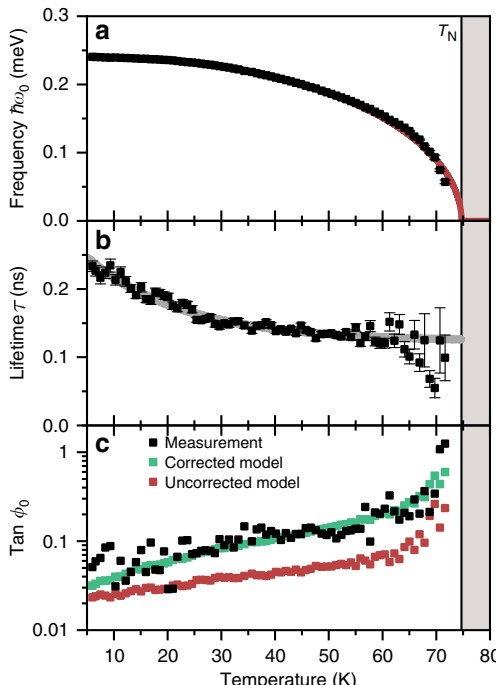

**Fig. 4 Temperature-dependent Z-mode characterisation in YMnO₃.** **a** Frequency, **b** relaxation time, and **c** tangent of the initial phase of the optically induced spin precession as function of temperature. Green and red: model calculations with and without correction for non-damping-based excitation mechanisms, respectively. Grey line: fitted exponential decay.

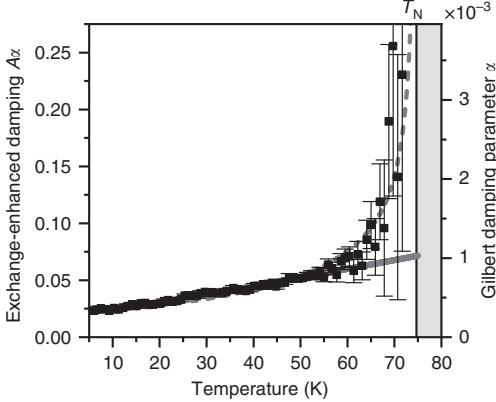

**Fig. 5 Temperature-dependent damping of the Z mode in YMnO₃.**
Gilbert damping parameter $\alpha$ and exchange-enhanced damping $A\alpha$ are shown in the same plot with $A = 69.3$ (see Methods section, Table 1). Full line: offset Bose-Einstein distribution with constant quasiparticle energy $E = 2.6$ meV. Dashed line: offset Bose-Einstein distribution with $E \propto \omega_0$ and $E(0\,\text{K}) = 2.6$ meV.

excitation mechanisms, such as an optical spin transfer torque[35,36] or optical orientation[37], which could become relevant because of the smallness of $\alpha$ in YMnO₃ as compared to HoMnO₃. These mechanisms, which are based on absorption rather than spin damping, involve an angular momentum transfer from the circularly polarised pump pulse to the material. As the angular momentum of the circular laser pulse is directed along the z axis, we heuristically include such a contribution in our model by setting $M_{i,z}(+0) = \alpha\gamma\mu_0H\theta M_0 + \delta$ in Eq. (3), which changes the initial phase to

$$\tan\phi_0 \approx A\left(\alpha + \frac{\delta}{\gamma\mu_0H\theta M_0}\right). \tag{8}$$

With this, we restore the quantitative agreement between the measured and modelled initial phase in Fig. 4c.

## Discussion

We thus demonstrated the indispensable role of the previously neglected ultrafast damping-like torque during the optical excitation of AFMs, both experimentally and theoretically. The torque impulsively generates a magnetisation along the minor axis of the intrinsically highly elliptical spin precession in AFMs. The extreme ellipticity converts the small damping-related magnetisation into a large precessional canting. Because of this leverage effect, this damping-based mechanism can even facilitate the dominant spin excitation mechanism in striking contrast to its complete neglect thus far. In addition to studying the spin dynamics deep in the antiferromagnetic phase, we also traced the associated Z mode in the vicinity of both first- and second-order phase transitions. This enabled us to verify the role of the ultrafast damping-like torque in distinctly different scenarios. We find that the Gilbert damping parameter $\alpha$ remains almost constant throughout the SRT in HoMnO₃, while the changes of the magneto-crystalline anisotropy driving the SRT lead to a twofold increase in the exchange enhancement factor $A$. In contrast, as we approach the Néel temperature $T_N$ (here in YMnO₃), the magneto-crystalline anisotropy remains unchanged, but $\alpha$ diverges. Although fundamentally different, we find an enhanced ultrafast damping-like torque in both cases, which again highlights the general importance of this hitherto neglected excitation mechanism.

More specifically, in terms of damping mechanisms we identified scattering with the $4f$ crystal-field excitations of HoMnO₃ as the leading contribution to the spin damping. Therefore, the choice of the A-site ion enables us to tune the strength of the damping across several orders of magnitude. This is, on the one hand, a useful degree of freedom for tuning material properties to specific spintronics applications. On the other hand, it allowed us to identify the presence of additional non-damping-based mechanisms inducing a net magnetisation. With respect to this, we note that any ultrafast mechanism that induces a net magnetisation in an AFM will benefit from the observed exchange enhancement. While field-like excitations via the IFE or using THz pulses typically induce spin-cantings of the order of 1°[15,38], there are various possibilities beyond increasing the pump fluence that may allow us to exceed a 90° spin canting and thus induce antiferromagnetic switching. Exploiting different excitation mechanisms such as thermally and optically induced ultrafast spin-transfer torques[35,39,40] may drastically increase the impulsively induced net magnetisation. Increasing the exchange enhancement, e.g. by tuning the system close to a SRT, will increase the ellipticity of the spin precession. Therefore, the impulsively induced net magnetisation translates into a larger modulation of the antiferromagnetic order parameter thereby reducing the switching threshold. Ultimately, our results show that both field-like and damping-like torques – a combination that has proved to be powerful for the electrical control of magnetic order[7,41] – are available optically, too, and can be utilised to act on the magnetic order. Thus, optically generated torques might provide the long sought-after tool enabling the efficient realisation of ultrafast coherent precessional switching of AFMs.

## Methods

**Setup**. Both samples are flux-grown and polished (0001)-oriented platelets of ~55 μm thickness. Our setup is schematically shown in Fig. 1b. The fundamental light source is a regeneratively amplified laser system providing 130 fs pulses at 1.55 eV photon energy and 1 kHz repetition rate. We use the output of an optical parametric amplifier combined with a quarter-wave plate to generate circularly polarised pump pulses at 0.95 eV photon energy, which create an effective magnetic field in the sample via the IFE. The pump fluence on the sample is ~60 mJ cm⁻². By pumping the material far below the nearest absorption resonance at ≈1.6 eV[42], we minimise parasitic heating effects[43]. We then probe the ensuing dynamics by measuring the time-resolved Faraday rotation $\Phi(t)$ of a linearly polarised probe pulse at 1.55 eV with balanced photodetectors. The probe fluence is <0.1 mJ cm⁻². Both the pump and probe beams are arranged in a quasi-collinear geometry with the beam propagation direction along the hexagonal z axis, which is the optic axis. Thus, the incident light experiences optical isotropy and complications arising from birefringence or other anisotropic optical effects are avoided. The sample is mounted in a cryostat for varying the temperature between ≈5 K and 300 K.

The oscillatory part of the signal (Fig. 2a) is fitted by

$$\Phi(t) = A_0 + A_1 e^{-t/t_1} + A_2 e^{-t/\tau}\sin(\omega t + \phi_0). \tag{9}$$

While the first two terms model parasitic contributions to the signal arising from thermal electron dynamics, we extract the frequency $\omega(T)$, relaxation time $\tau(T)$, and initial phase $\phi_0(T)$ from the oscillatory third term. Error bars in Figs. (3), (4), and (5) reflect the standard error of that fit or its propagation through Eqs. (4)–(6).

**Temperature dependence of $\omega_0$**. As stated in the main text, the analytic solution of the Landau-Lifshitz-Gilbert equation yields (Supplementary Note 1): $\hbar\omega_0 = 2g\mu_B M_0\sqrt{\mathcal{J}\mathcal{D}}$. According to mean-field theory, the temperature-dependent sublattice magnetisation $M_0(T)$ can be described as a paramagnet aligned by the exchange field of the neighbouring sublattices, i.e. $M_0(T) = Ng\mu_B S\mathcal{B}_{S=2}(M_0(T)T_N T^{-1})$, where $\mathcal{B}_{S=2}$ is the Brillouin function of an $S = 2$ Mn spin[16,44,45] and $N = 6/V$ with $V = 0.374$ nm³, being the unit cell volume, is the number density of Mn atoms. We model the SRT in HoMnO₃ by introducing a linear dependence of the anisotropy parameter $D_y$ on temperature. The SRT temperature $T_{SR}$ is defined as the point where $D_y$ vanishes[24], i.e. $D_y(T) = \kappa(T - T_{SR})$. For YMnO₃, assuming a temperature-independent value for $D_y$ yields a reasonable fit. This model is fitted to the measured temperature dependence of $\omega_0$. The results are summarised in Table 1.

**Table 1 Fit parameters.** Values for $\lambda = \frac{J}{N(g\mu_B)^2}$ and $D_z = \frac{K_z}{N(g\mu_B)^2}$ were calculated from the exchange interaction $J$ and the out-of-plane anisotropy constant $K_z$ given in the respective references.

| Parameter | Unit | HoMnO$_3$ | YMnO$_3$ |
|---|---|---|---|
| $T_N$ | [K] | 69.2(7) | 74.7(1) |
| $T_{SR}$ | [K] | 33.5(2) | – |
| $\lambda$ | [T$^2$ cm$^3$ J$^{-1}$] | 70.8[46] | 71.1[25] |
| $D_z$ | [T$^2$ cm$^3$ J$^{-1}$] | 11.0[46] | 13.9[25] |
| $D_y(T)$ | [T$^2$ cm$^3$ J$^{-1}$] | $\kappa(T_{SR} - T)$ | 0.0251(1) |
| $\kappa$ | [T$^2$ cm$^3$ J$^{-1}$ K$^{-1}$] | 0.0093(6) | 0 |
| $\Delta \cdot (g\mu_B)^2$ | [T$^2$ cm$^3$ J$^{-1}$] | 0.0354(9) | 0 |
| $\mathcal{J}$ | [T$^2$ cm$^3$ J$^{-1}$] | $117.2 + |D_y|$ | 120.6 |
| $\mathcal{D}$ | [T$^2$ cm$^3$ J$^{-1}$] | $\Delta M_0^2 + |D_y|$ | 0.0251(1) |
| $A = \sqrt{\mathcal{J}/\mathcal{D}}$ | [1] | Fig. 3e | 69.3 |

## Data availability

The data that support the findings of this study are available from the corresponding author on reasonable request. Source data are provided with this paper.

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

## Acknowledgements

The authors thank B.A. Ivanov and A.V. Kimel for valuable discussions and R. Gmünder for experimental assistance. T.S. was supported by the Japan Society for the Promotion of Science (JSPS) KAKENHI (Nos. JP19H01828, JP19H05618, JP19K21854, and JP26103004) and the JSPS Core-to-Core Program (A. Advanced Research Networks). C.T. acknowledges support by the SNSF under fellowship P2EZP2-191801. C.T. and M.F. acknowledge support from the SNSF project 200021/147080 and by FAST, a division of the SNSF NCCR MUST.

## Author contributions

C.T. and T.S. conceived the project. C.T. designed and evaluated the experiments. T.S. and M.F. supervised the project. All authors discussed the results and contributed to writing the manuscript.

## Competing interests

The authors declare no competing interests.
