## [Peer Review File · Nature Communications]

Reviewers' Comments:

Reviewer #1:

Remarks to the Author:

The authors answered all comment of the referees in detail and added changes to the manuscript accordingly. I therefore I support a publication in Nature Communications.

Reviewer #2:

Remarks to the Author:

The authors have responded adequately to nearly all of my comments and I therefore find this manuscript suitable for publication. However, the comment in my initial review that this work is rather specialized and only of interest to a rather limited community (a small subset of those interested in ultrafast dynamics in antiferromagnetic materials) have not been addressed or refuted. I therefore am of the opinion that this paper is not of sufficient general interest to the readers of Nature Physics and is more appropriate for a more specialized journal.

Responses to reviewers' comments

Reviewer #1:

The authors answered all comment of the referees in detail and added changes to the manuscript accordingly. I therefore I support a publication in Nature Communications.

We are deeply grateful for the reviewer's recommendation to publish our manuscript.

Reviewer #2:

The authors have responded adequately to nearly all of my comments and I therefore find this manuscript suitable for publication. However, the comment in my initial review that this work is rather specialized and only of interest to a rather limited community (a small subset of those interested in ultrafast dynamics in antiferromagnetic materials) have not been addressed or refuted. I therefore am of the opinion that this paper is not of sufficient general interest to the readers of Nature Physics and is more appropriate for a more specialized journal.

We kindly thank the reviewer for finding our manuscript suitable for publication.

We do, however, not share the reviewer's view regarding the generality of our results. We show that the damping-like torque, which was so far neglected for ultrafast optical excitations, can provide the dominating contribution to coherent spin excitations. While we demonstrated the consequences of the damping-like torque specifically for spin excitations via the inverse Faraday effect, our findings have implications for all mechanisms of coherent spin excitation. The relevance of the damping-like torque thus extends beyond the inverse Faraday effect to spin excitations using e.g. THz radiation or optically generated spin transfer torques.

Moreover, the versatility provided by the availability of field-like and damping-like torques has proved to be a powerful tool to manipulate magnetic order electrically in the context of spintronics. Our results show that both torques are available optically, too, and can be utilized to act on the magnetic order. We therefore believe that our manuscript will not only be appreciated by the ultrafast antiferromagnetic dynamics community, but also by the broader spintronics community.

We now highlight this general result in lines 171-173 in the main text.

Despite our disagreement we acknowledge that there can be different opinions and therefore transferred the manuscript to Nature Communications, which the reviewer might have missed, as her/his reference to Nature Physics suggests.